# Simple and Efficient Protocol for Subcellular Fractionation of Normal and Apoptotic Cells

**DOI:** 10.3390/cells10040852

**Published:** 2021-04-09

**Authors:** Viacheslav V. Senichkin, Evgeniia A. Prokhorova, Boris Zhivotovsky, Gelina S. Kopeina

**Affiliations:** 1Faculty of Medicine, MV Lomonosov Moscow State University, 119991 Moscow, Russia; slsenichkin@gmail.com (V.V.S.); eugeniaprokhorova@gmail.com (E.A.P.); 2Sir William Dunn School of Pathology, University of Oxford, Oxford OX1 2JD, UK; 3Institute of Environmental Medicine, Karolinska Institutet, Box 210, 17177 Stockholm, Sweden

**Keywords:** apoptosis, cytosol, nuclei, fractionation, translocation

## Abstract

Subcellular fractionation approaches remain an indispensable tool among a large number of biochemical methods to facilitate the study of specific intracellular events and characterization of protein functions. During apoptosis, the best-known form of programmed cell death, numerous proteins are translocated into and from the nucleus. Therefore, suitable biochemical techniques for the subcellular fractionation of apoptotic cells are required. However, apoptotic bodies and cell fragments might contaminate the fractions upon using the standard protocols. Here, we compared different nucleus/cytoplasm fractionation methods and selected the best-suited approach for the separation of nuclear and cytoplasmic contents. The described methodology is based on stepwise lysis of cells and washing of the resulting nuclei using non-ionic detergents, such as NP-40. Next, we validated this approach for fractionation of cells treated with various apoptotic stimuli. Finally, we demonstrated that nuclear fraction could be further subdivided into nucleosolic and insoluble subfractions, which is crucial for the isolation and functional studies of various proteins. Altogether, we developed a method for simple and efficient nucleus/cytoplasm fractionation of both normal and apoptotic cells.

## 1. Introduction

Numerous biochemical changes occur in cells as a consequence of the translocation of proteins into and out of the nucleus. For example, dozens of studies have demonstrated the importance of nuclear import of proteins for transcription induction, while their export to the cytoplasm decreases the transcription of target genes [1]. Apoptosis is one of the physiological processes characterized by alterations in the localization of multiple proteins, including various examples of translocation of apoptotic regulators between different intracellular compartments [2]. Thus, upon apoptosis induction, cytochrome *c* translocates from the mitochondria to cytosol, while AIF, endonuclease G (EndoG), and HtrA2/Omi are also released from the intermembrane space of the mitochondria but then translocate into the nucleus [3,4]. Ca^2+^-dependent endonuclease DNAS1L3 is reported to relocate from the endoplasmic reticulum (ER) to the nucleus in a caspase-dependent manner [5]. Caspases themselves, which are the main players in apoptosis induction and execution, have been shown to translocate to the nucleus and cleave many of their nuclear substrates [6,7]. Hence, evaluation of subcellular localization of proteins is of great significance for studying their functional activity, including that of apoptosis regulators. 

For this purpose, either biochemical fractionation approaches or various microscopy techniques, in particular, confocal microscopy, can be applied [8]. Both methodologies have their pros and cons. Confocal microscopy allows for single-cell analysis, can be performed on living cells, and is appropriate for quantitative analysis. However, the use of confocal microscopy is significantly limited by the availability of equipment, which is not a problem for biochemical fractionation methods that require only widely used equipment for western blot analysis. Limited choice of high-quality primary antibodies is another factor that remains restrictive in confocal microscopy. Furthermore, in some cases, the amounts of protein in a fraction may simply be insufficient for its detection by microscopy. Additionally, dying cells usually detach from the plastic surface, which hampers the study of protein translocation in response to cytotoxic treatments. In contrast, since subcellular components obtained by methods of biochemical fractionation can be analyzed separately, it is feasible to amplify the signal specifically in the fraction of interest even if the protein concentration is low. Last but not least, the isolated components can be used in various downstream analyses, including western blot, immunoprecipitation, and mass-spectrometry, allowing for studying protein–protein interaction, different isoforms or truncations of proteins, and their post-translational modifications. Hence, biochemical fractionation techniques remain one of the main and most frequently used approaches for determining the localization of proteins and studying their functions.

Biochemical fractionation was first described as early as in the mid-20th century. Since that time, numerous variations of fractionation protocols have been developed; however, most are still based on the principles originally proposed in those early studies, i.e., on the use of non-ionic detergents [9,10], sucrose gradient [11], or different types of homogenizers [12]. At the same time, the finest details in fractionation protocols are particularly important for the improvement of fractionation quality, that is, the ability of the method to efficiently separate the cell contents into pure cytoplasmic and nuclear components. Another important characteristic of the protocol is its simplicity. Hence, the most appropriate nucleus/cytoplasm fractionation protocol should both be simple for application and allow for efficient isolation of proteins from subcellular fractions. 

Here, we analyzed various fractionation techniques: fractionation using a Potter-Elvehjem homogenizer, fractionation with non-ionic detergents (digitonin or NP-40), and stepwise lysis of cells and washing of the resulting nuclei using NP-40 at both stages. The latter approach uses a similar principle of successive lysis and washing as previously described by the REAP (Rapid, Efficient and Practical) method [9], but includes several important modifications. Thus, in contrast to this method’s original design, we used: (1) hypotonic solution before lysis of cell membrane, (2) longer time intervals at some stages, and (3) various concentrations of non-ionic detergent in washing solution depending on the cell line. Based on the principle of the used approach, we designate it as “L&W” (“Lyse-and-Wash”). The quality of the tested approaches was controlled by western blot by staining for specific markers of the cell membrane, ER, mitochondria, and nucleus. Additionally, confocal microscopy was used to assess the purity of the isolated nuclei. In contrast to other fractionation techniques used in this work, the L&W method has demonstrated excellent results for the isolation of nuclear proteins with no markers from other compartments detected by western blot. Moreover, using confocal microscopy, it was shown that L&W allows for isolation of pure nuclei devoid of ER remnants without disintegration of nuclear morphology.

Consistently, this approach was selected for further validation in cells treated with various apoptotic stimuli. The demolition of the nucleus is one of the characteristic features of apoptosis [13], and it can be associated with the disruption of the integrity of the nuclear envelope and redistribution of proteins between the cytoplasm and the nucleus. Hence, one of the potential problems of biochemical fractionation methods in the context of apoptotic cells might be the issue of effective separation of cytoplasmic and nuclear proteins. Moreover, apoptotic bodies, as well as fragments of apoptotic cells, can contaminate the nuclear fraction leading to incorrect assessment of protein localization. Nevertheless, the L&W method has demonstrated accurate isolation of nuclear fraction even from cells treated with various apoptotic stimuli. 

Finally, using an approach developed by us, we demonstrated that the nuclear fraction can be further divided into RIPA-soluble and -insoluble subfractions. The former contains nucleosolic proteins, while the latter is enriched with proteins tightly associated with the insoluble part of the nucleus, including DNA and nuclear membranes. We show that several proteins were specifically detected in RIPA-insoluble fraction, indicating their association with DNA or nuclear envelope. Thus, the described approach can be used to separate various subsets of nuclear proteins.

Altogether, in this work, we describe a highly effective method for nucleus/cytoplasm fractionation of both living and apoptotic cells that will facilitate functional studies of nuclear events and protein translocation, in particular, in cells undergoing apoptosis.

## 2. Materials and Methods

### 2.1. Cell Lines and Cultivation

Ovarian adenocarcinoma cells Caov-4 (ATCC HTB-76) and cervical adenocarcinoma cells HeLa (ATCC CCL-13) were kindly provided by the Department of Toxicology, Karolinska Institutet (Stockholm, Sweden). The cells were cultured in DMEM high glucose medium (Gibco, Paisley, Scotland, UK) supplemented by 10% fetal bovine serum (Gibco), 1 mM sodium pyruvate (PanEco, Moscow, Russia), penicillin (100 U/mL), and streptomycin (100 μg/mL) (Gibco). Cells were grown in a CO_2_ incubator (5% CO_2_) at 37 °C. For collection, cells were washed with Versene solution (PanEco) and then incubated with 0.15% trypsin solution for 2 min (Gibco). Cell death induction experiments were performed when cells were in the logarithmic growth phase. For cell death induction, 0.1 µM staurosporine (Sigma Aldrich, St. Louis, MO, USA), 10 ng/mL TNF-α (Generium, Moscow, Russia) + 5 μg/mL cycloheximide (Sigma), or 35 µM cisplatin (Teva, Yaroslavl, Russia) were used.

### 2.2. Antibodies

The following antibodies were used for western blot analysis: GAPDH (#2118), H2AX (#2595), Na/K-ATPase (#3010), caspase-3 (#9662) (all from Cell Signaling Technology, Beverly, MA, USA); caspase-2 (sc-5292), PARP-1 (sc-7150), Lamin B1 (sc-374015) (all from Santa Cruz Biotechnology, Santa Cruz, CA, USA); cytochrome *c* (#556432, BD Biosciences, Franklin Lakes, NJ, USA); caspase-8 (5F7, Enzo Life Sciences, Farmingdale, NY, USA); caspase-9 (MA1-16842, Thermo Scientific, Rockford, IL, USA); and rabbit anti-ERp29 (kindly provided by Dr. S. Mkrtchian, Karolinska Institutet). HRP-linked goat anti-rabbit and HRP-linked goat anti-mouse (#97200 and #97265, respectively; Abcam, Cambridge, MA, USA) were used as secondary antibodies. 

### 2.3. Nucleus/Cytoplasm Fractionation

All preparations were performed on ice. Cells were resuspended in 1 mL hypotonic solution containing 0.1% NP-40 and incubated for 3 min. Next, cells were homogenized using a Potter-Elvehjem homogenizer by ~20 iterations of up and down passes of the pestle. The solution was centrifuged to pellet nuclei (1000 rcf, 5 min). Supernatant (cytoplasmic fraction) was re-centrifuged (15,000 rcf, 3 min) to pellet debris. Fractionation with non-ionic detergents was carried out by adding a hypotonic solution to the cells for 3 min. Then, NP-40 or digitonin were added to a final concentration 0.1%. The resulting solutions were kept for 3 min and centrifuged (1000 rcf, 5 min). Supernatant (cytoplasmic fraction) was re-centrifuged (15,000 rcf, 3 min) to sediment debris.

Fractionation by the L&W method and its variations (including those with DNase I addition and subfractionation using RIPA-buffer) is summarized as a step-by-step protocol in the section “The L&W nucleus/cytoplasm fractionation protocol.” The compositions of the hypotonic, isotonic, DNase I, and RIPA buffers are also given in the same section.

### 2.4. Western Blot Analysis

The total cell lysate was obtained by lysing cells in RIPA buffer. To prepare samples for western blot analysis, 30 μg of protein were taken from cell lysates or cytoplasmic fraction solutions and mixed with Laemmli buffer. Afterward, the volumes in all samples were adjusted. To prepare samples with nuclear proteins, Laemmli buffer was added to nuclear pellets obtained by different fractionation methods. Samples (cellular, cytoplasmic, or nuclear) were then heated at 95 °C for 5 min. Afterward, the samples were separated by PAGE (4% stacking gel, 12% resolving gel) and transferred to a nitrocellulose membrane (Bio-Rad, Hercules, CA, USA) using Mini Trans-Blot cells (Bio-Rad). The membranes were blocked for 40 min in milk (5% solution in TBS) and stained sequentially with primary and secondary antibodies. To obtain the signal, the membranes were treated with ECL Western Blotting Substrate (Promega, Madison, WI, USA) or SuperSignal West Dura Extended Duration Substrate (Thermo Scientific) on a Molecular Imager ChemiDoc (Bio-Rad). If staining with additional antibodies was required, the membranes were incubated in a Restore Western Blot Stripping Buffer (Thermo Scientific) for up to 15 min, washed with TBS solution, blocked with milk, and the staining was repeated.

### 2.5. Confocal Microscopy

The cell/cell nucleus pellets were dissolved in PBS. Next, DAPI and ER-Tracker™ Green (BODIPY^®^ FL Glibenclamide; Invitrogen, Life Technologies Ltd., Paisley, Scotland) were added to the resulting suspension (to a final concentration of 1 µg/mL and 1 µM, respectively). After 10 min of incubation in a dark place, the suspension was centrifuged. The resulting pellet was washed in PBS solution and centrifuged again. Then, the pellet was redissolved in PBS, and a small amount of the suspension was spotted onto a glass slide into the antifade mounting medium Vectashield (Vector Laboratories, Burlingame, CA, USA). The microscope slides were analyzed using an LSM 780 confocal laser scanner microscope (Zeiss, Jena, Germany). Images were processed using ZEN software (Zeiss).

## 3. Results

### 3.1. Incubation in Hypotonic Buffer Increases the Quality of Fractionation

Two cancer cell lines were used for validation of fractionation protocol, ovarian adenocarcinoma cells Caov-4 and cervical adenocarcinoma cells HeLa. To assess the efficiency of separation of nuclear and cytoplasmic fractions, western blot analysis and confocal microscopy were applied. In western blot analysis, the purity of the nuclear fraction was assessed by staining for specific markers of the cell membrane (Na/K ATPase), ER (ERp-29), mitochondria (cytochrome *c*), and cytosol (GAPDH). Lamin B1 was used as a nuclear envelope marker, and H2AX was used as a nucleoplasm marker. In confocal microscopy, staining nuclei with DAPI and ER-Tracker dye for the detection of ER remnants was performed.

We used the principle of stepwise lysis and washing with non-ionic detergent solutions for the development of the protocol, which was called “L&W” (“Lyse-and-Wash”). In contrast to other protocols, cells were incubated in hypotonic solution before lysis. Additionally, we tested different times of protocol step duration and selected optimal (for details, see the section “The L&W nucleus/cytoplasm fractionation protocol”). Both changes were essential to increase the quality of fractionation, as shown below.

First, the L&W method and its modification without incubation of cells in hypotonic solution were tested. Western blot analysis demonstrated that L&W was efficient irrespective of incubation of cells in a hypotonic solution, as no cytoplasmic markers were detected in the nuclear samples and vice versa (Figure 1A). However, when confocal microscopy was used to evaluate the purity of isolated nuclei, a dramatic difference was observed between the samples obtained by these approaches. Indeed, nuclei isolated by L&W without the hypotonic solution step were positively stained for ER remnants, while the use of hypotonic solution before lysis enabled to effectively discard the ER compartment (Figure 1B). The difference in the results obtained by western blot analysis and confocal microscopy is explained by the fact that ERp-29 localizes to the ER lumen [14], while the sulfonylurea receptor stained by ER-Tracker™ Green (BODIPY^®^ FL Glibenclamide) is an ER membrane protein. Thus, non-ionic detergents effectively permeabilize the ER membrane but cannot efficiently dissolve it. In contrast, the use of hypotonic solution leads to the swelling of ER, which alleviates its dissolution during the washing steps and allows the isolation of pure, intact nuclei. 

We also demonstrated that shorter time intervals affect the purification of nuclei from ER. When samples were prepared with the duration of steps corresponding to the original design of the REAP protocol [9], ER was stained as assessed by confocal microscopy (Figure 1B). In contrast, using longer periods of lysis, washing, and centrifugation steps (see the protocol of L&W) resulted in total purification of nuclei from the ER compartment (Figure 1B). 

Finally, 0.1% NP-40 in washing solution was needed to obtain pure nuclei from Caov-4 cells. However, 0.3% NP-40 was required to obtain nuclei completely devoid of ER from HeLa cells (Figure 1C). Indeed, using higher light exposure in confocal microscopy experiments, we did observe low amounts of ER in HeLa nuclear fraction washed with 0.1% NP-40, whereas the increasing concentration of NP-40 in the washing solution improved the purity of nuclear samples (Figure 1C). Hence, the adjustment of the concentration of non-ionic detergent in washing solution might be required for better purification of nuclei. It is likely that larger cells (such as HeLa in our experiments) will require higher concentrations of detergent to efficiently remove the ER compartment. However, as 0.1% NP-40 was sufficient to almost completely discard ER from nuclei, this concentration was used in the L&W method for both cell lines in the following sections.

### 3.2. The Washing Step Is Indispensable for the Purification of Nuclear Fraction in the L&W Method

Next, we compared the described approach with other widely used fractionation methods. For this purpose, fractionation with non-ionic detergents (digitonin or NP-40) and fractionation using Potter-Elvehjem homogenizer were tested. Of note, all the methods included incubation in hypotonic buffer at the first stage.

Western blot analysis demonstrated that the L&W method ensured accurate separation of nuclear and cytoplasmic fractions in both tested cell lines as assessed by western blot (Figure 2A). The addition of DNase I to nuclear fractions, in order to reduce their viscosity, did not affect the quality of fractionation. Therefore, DNase I or other endonucleases can be used to optimize sample preparation and gel loading. In contrast to L&W, other tested approaches showed rather low purity of nuclear isolation. Thus, nuclear fractions obtained by using non-ionic detergents, digitonin, and NP-40 were contaminated by ER, mitochondrial, and cytoplasmic markers. Moreover, low levels of Na/K ATPase were detected in cytoplasmic samples after fractionation by digitonin, which indicates relatively low solubilization of membrane proteins by this compound. Finally, the use of a Potter-Elvehjem homogenizer was relatively efficient for fractionation of Caov-4 but not of HeLa cells, which might be due to differences in the size of these cells. Additionally, no nuclear envelope marker, Lamin B1, was found in Caov-4 cell nuclear fraction, which means that this type of fractionation can significantly disrupt the integrity of the nuclear membrane (Figure 2A).

The efficiency of the tested fractionation techniques was also verified by confocal microscopy. Fractionation using non-ionic detergents was inefficient for the separation of the ER compartment from the nuclei in both tested cell lines (Figure 2B). As for the Potter-Elvehjem homogenization technique, this approach enabled the better separation of these compartments. However, the nuclei were significantly damaged by this approach (Figure 2B). In contrast, the L&W method provided accurate and efficient purification of nuclei, as no ER compartment was detected in both tested cell lines (Figure 2B), which is consistent with the data described above (Figure 1B). Hence, among the tested approaches, only L&W efficiently separated nuclear and cytoplasmic fractions. Of note, fractionation using NP-40 is similar to the L&W approach without the washing step with a non-ionic detergent. Washing of cells after lysis dramatically increased fractionation quality, which means that the washing step is crucial for efficient purification of the nuclear fraction.


### 3.3. The L&W Approach Allows Efficient Fractionation of Apoptotic Cells

As the L&W method demonstrated the highest quality of separation of nuclear and cytoplasmic proteins, this approach was further validated for fractionation of cells treated with various apoptotic stimuli, including stimulation of both extrinsic and intrinsic apoptotic pathways. For apoptosis induction, Caov-4 and HeLa cells were treated with promiscuous kinase inhibitor staurosporine (0.1 µM), a combination of TNF-α (10 ng/mL) and protein synthesis inhibitor cycloheximide (5 μg/mL)—stimulator of the extrinsic pathway—and DNA-damaging agent cisplatin (35 µM)—activator of the intrinsic pathway of cell death. Next, cells were fractionated using the L&W approach, and the resulting cytoplasmic and nuclear fraction samples were analyzed by western blot. The purity of the samples was assessed by staining for Na/K ATPase, ERp-29, cytochrome *c*, Lamin B1, and H2AX (Figure 3). PARP-1 staining was performed to evaluate the induction of apoptotic cell death. This protein is a well-known target of activated caspases, in particular of executioner caspase-3, the main effector of apoptosis, and cleavage of full-length PARP-1 to the 89 kDa fragment is a hallmark of apoptotic cell death [15]. In addition to markers of the above-mentioned intracellular compartments, samples were also stained for several members of the caspase family, namely, initiator caspase-2, -8, -9, and executioner caspase-3. A number of studies have shown that apoptosis induction is accompanied by the translocation of these proteins into the nucleus, where they implement the apoptotic program [2,6,7]. 

It was shown that in Caov-4 and HeLa cells, apoptosis induction did not affect fractionation efficiency. The induction of apoptosis was confirmed by PARP-1 cleavage and generation of its p89 fragment, as well as by generation of active forms of caspases. Therefore, the L&W method has demonstrated its applicability for the subcellular fractionation of apoptotic cells. Using western blot, we also demonstrated that at least low levels of activated caspases could be detected in nuclear fractions of cells treated with three different apoptotic stimuli. Thus, we detected the translocation into the nucleus of caspase-2, -3, -8, and -9 in Caov-4 cells treated with TNF-α + cycloheximide; of caspase-9 in Caov-4 cells treated with cisplatin and of caspase-2, -8, and -9 in HeLa cells treated with either of the tested apoptotic inducers (Figure 3). There were no caspases detected in the nuclei of untreated cells. Taken together, caspases can translocate to the nuclei during the execution of the apoptotic program to facilitate the disruption of this compartment.


### 3.4. The Nuclear Fraction Can Be Subdivided into Two Fractions: Soluble and Insoluble

In the nucleus, proteins can be present in the nucleosol or strongly associated with insoluble components, in particular chromatin and the nuclear membrane. Consequently, the low yield of proteins after fractionation may be not only because of their low levels in the nucleus but also due to their poor isolation from the insoluble nuclear fraction. To prove this hypothesis, the soluble and insoluble subfractions of the nuclei were analyzed for the presence of caspases. For this purpose, the extraction protocol described below in the section “The L&W nucleus/cytoplasm fractionation protocol” was utilized. In brief, this protocol is based on the solubilization of proteins from a nuclear pellet using RIPA buffer. RIPA-soluble portion comprises nucleosolic proteins, while the RIPA-insoluble fraction predominantly contains proteins associated with chromatin and the nuclear envelope.

Caov-4 and HeLa cells were treated with cisplatin. The presence of caspases at various time points in both RIPA-soluble and -insoluble fractions were assessed. Besides procaspases (i.e., full-length caspases), which are catalytically inactive zymogens, their cleaved products representing active enzymes were analyzed. Strikingly, both procaspases and processed caspases were found in insoluble nuclear fractions but not soluble ones (Figure 4). These results were similar for both cell lines. Hence, caspases do not localize to the nucleosol but rather associate with various substrates that are tightly bound to DNA, nuclear envelope, or lamina proteins. In support of this assumption, a caspase substrate PARP-1 was also enriched in RIPA-insoluble fractions.

Surprisingly, GAPDH has also been detected in RIPA-insoluble fraction. At the same time, there are some other studies that demonstrate the capability of this protein to translocate to the nucleus following genotoxic stress [16,17]. Hence, these results should not be considered as an artifact but rather reflect the multifunctionality of GAPDH.

## 4. Discussion

Here, we describe a simple and efficient nucleus/cytoplasm fractionation approach, which is based on the lysis of the cytoplasmic membrane and subsequent washing of the nuclei with non-ionic detergent. The key stage of this method was the use of a washing step, which was previously proposed by the authors of the REAP method [9]. Indeed, we show that fractionation with NP-40 without the washing step was ineffective for the separation of nuclei from ER and mitochondria, while the addition of the washing step resulted in a significant increase in the quality of fractionation. However, our L&W protocol differs from REAP in several important aspects: we used incubation of cells in a hypotonic solution at the first stage and longer duration of protocol steps, which both independently resulted in extremely better purification of nuclear fraction from ER compartment, as evidenced by confocal microscopy. 

We also show that the concentration of non-ionic detergent in a washing solution can vary depending on both the cell type and the requirements for the purity of nuclear fraction (thus, if low content of ER compartment in the nuclear fraction is acceptable, the concentration can be lower). Therefore, some details of the protocol might be adjusted if necessary. It should also be noted that other non-ionic detergents similar to NP-40 (such as Triton X-100 and IGEPAL CA-630) can be used in the described protocol. However, the use of other (i.e., not similar to NP-40) non-ionic detergents might affect the quality of fractionation. Thus, the use of digitonin, which is a milder non-ionic detergent than NP-40, results in lower efficiency of fractionation, in particular in the context of removal of ER remnants.

Finally, we show that several proteins can be distributed unequally in nucleosolic and insoluble nuclear subfractions. In this case, buffers containing ionic detergents, such as RIPA (contains SDS), can be used to disrupt nuclear membranes and extract proteins of the soluble nuclear subfraction, i.e., nucleosolic proteins. On the other hand, RIPA-insoluble fraction comprises non-nucleosolic proteins and can be represented, in particular, by DNA-interacting proteins. Subfractionation of the nuclear fraction into RIPA-soluble and -insoluble portions might be useful for studying the roles of individual proteins in various biochemical processes.

In summary, we have provided the protocol for rapid and efficient nucleus/cytoplasm fractionation of both living and apoptotic cells (for step-by-step graphical representation and explanation, see Figure 5 and the Appendix A). This protocol will find a wide application in experimental investigations of protein cytoplasmic and nuclear functions and dynamics of protein redistribution in response to apoptotic and possibly other stress stimuli that affect cellular integrity.


## 5. Conclusions

In this report, we provide a comparison of several fractionation methods and describe a simple and efficient method for effective fractionation of cells. We conduct a comprehensive evaluation of the effectiveness of our approach, demonstrating the purity of the obtained fractions by confocal microscopy and western blot analysis. Our approach to obtaining cleaner nuclei and subnuclear fractions will also facilitate biochemical investigation of other nuclear processes where rigorous exclusion of organelle contamination, such as the ER, is necessary. Thus, we provide a solution for fractionation of both living and apoptotic cells and we are confident that it will save researchers time when choosing an effective method for their experiments.

## Figures and Tables

**Figure 1 cells-10-00852-f001:**
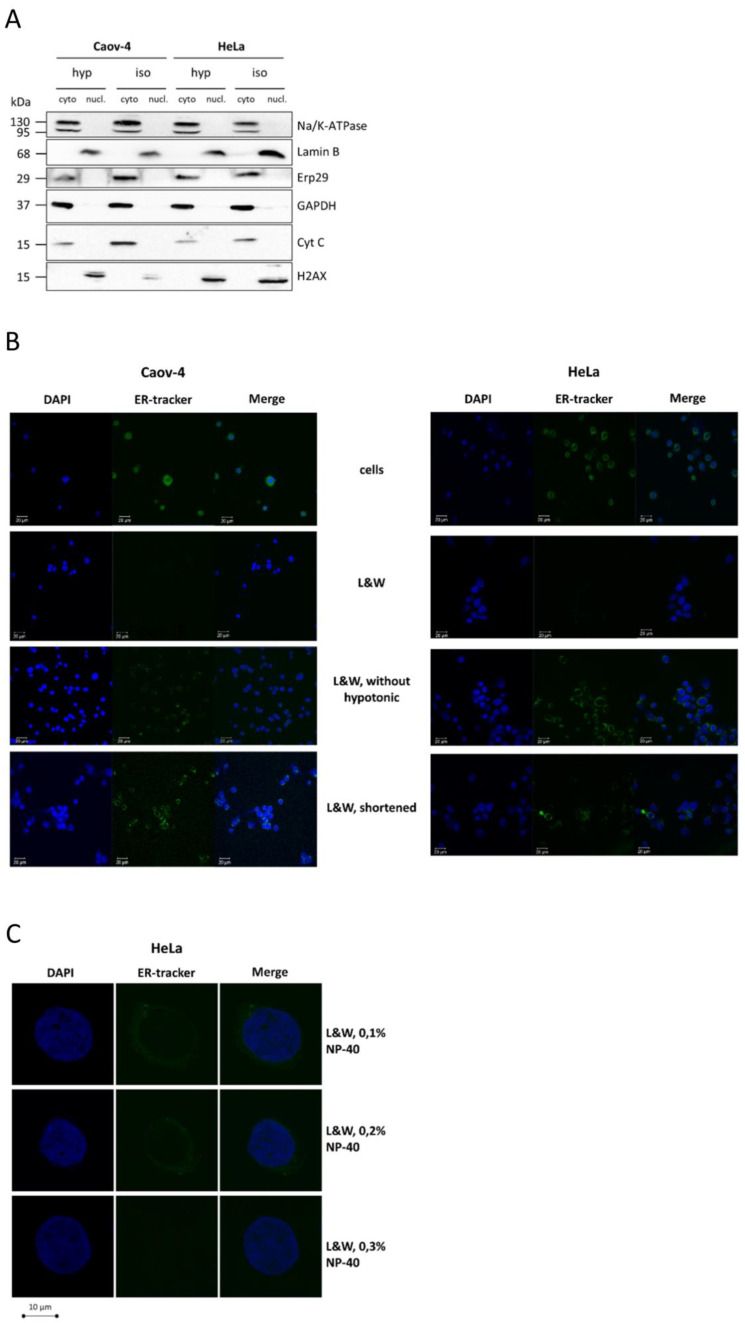
Optimization of the Lyse-and-Wash (L&W) nucleus/cytoplasm fractionation protocol. (**A**) Western Blot analysis of cytoplasmic and nuclear fractions from Caov-4 and HeLa cells obtained by the L&W approach with (hypo) or without (iso) pre-incubation of cells in a hypotonic buffer before lysis. Markers of the cell membrane (Na/K ATPase), ER (ERp-29), mitochondria (cyTable 2. AX) were used to control fractionation quality; (**B**) confocal microscopy of nuclei from Caov-4 and HeLa cells isolated by the L&W approach and its modifications, without pre-incubation of cells in hypotonic buffer or with shortened steps. The samples were stained with DAPI and ER-tracker; (**C**) confocal microscopy of nuclei from HeLa cells isolated using the L&W method with various concentrations of non-ionic detergent (NP-40, concentrations are indicated) in the washing solution. The samples were stained with DAPI and ER-tracker. High light exposure was used for this experiment. L&W, Lyse-and-Wash method.

**Figure 2 cells-10-00852-f002:**
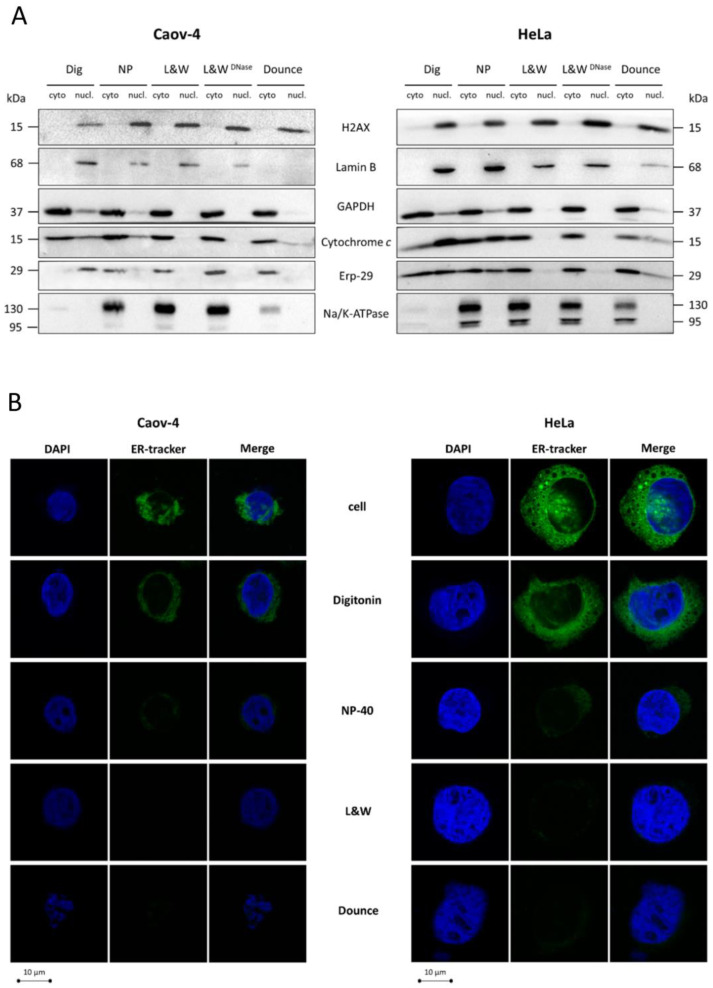
Comparative analysis of various biochemical nucleus/cytoplasm fractionation methods. (**A**) WB analysis of cytoplasmic and nuclear fractions from Caov-4 and HeLa cells obtained by various fractionation approaches. To assess the purity of the resulting fractions, samples were stained for markers of the cell membrane (Na/K ATPase), ER (ERp-29), mitochondria (cytochrome *c*), cytoplasm (GAPDH), nuclear envelope (Lamin B), and nucleoplasm (H2AX). Cyto, cytoplasmic fraction; nucl, nuclear fraction; (**B**) confocal microscopy of nuclei from Caov-4 and HeLa cells isolated by the tested fractionation methods. The samples were stained with DAPI and ER-tracker. L&W, Lyse-and-Wash method; 0.1% of NP-40 in the washing solution.

**Figure 3 cells-10-00852-f003:**
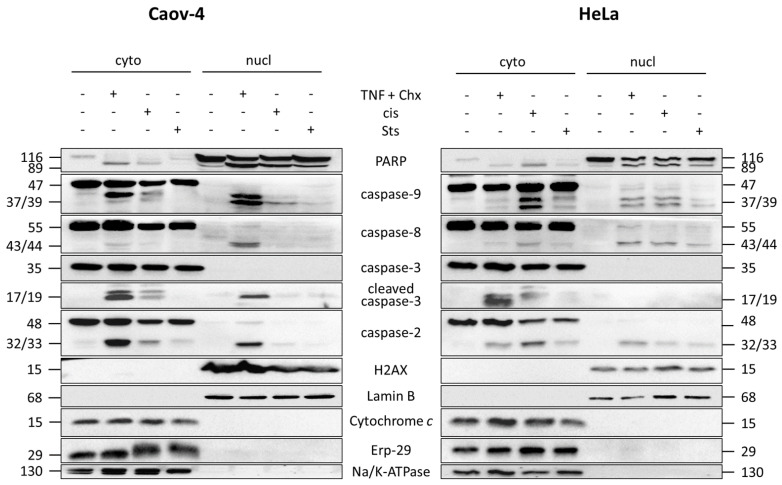
Validation of the L&W protocol for nucleus/cytoplasm fractionation of apoptotic cells. WB analysis of cytoplasmic and nuclear fractions from Caov-4 and HeLa cells treated with 0.1 µM staurosporine (Sts), 10 ng/mL TNF-α + 5 μg/mL cycloheximide (TNF + Chx), or 35 µM cisplatin (cis). The purity of the resulting fractions was assessed by staining for markers of the cell membrane (Na/K ATPase), ER (ERp-29), mitochondria (cytochrome *c*), nuclear envelope (Lamin B), and nucleoplasm (H2AX). PARP cleavage was evaluated as a marker of cell death. Cyto, cytoplasmic fraction; nucl, nuclear fraction.

**Figure 4 cells-10-00852-f004:**
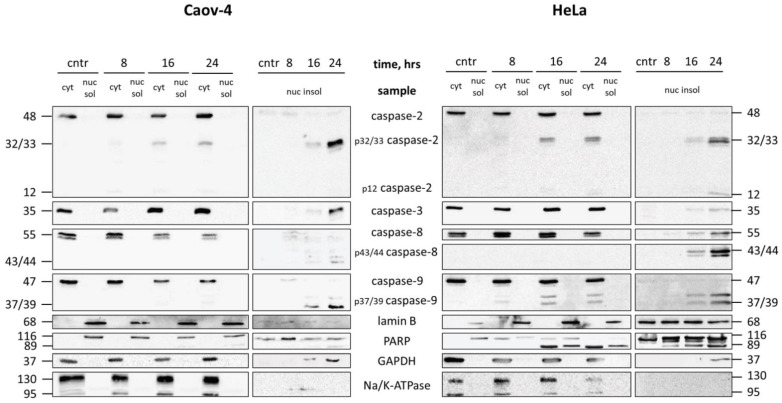
Analysis of RIPA-soluble and RIPA-insoluble portions of the nuclear fraction. WB analysis of cytoplasmic (cyt), RIPA-soluble (nuc sol), and RIPA-insoluble (nuc insol) fractions obtained from Caov-4 and HeLa cells treated with 35 µM cisplatin for the indicated time periods (hrs—hours). Cntr, control (untreated) cells. The purity of the resulting fractions was assessed by staining for markers of the cell membrane (Na/K ATPase), cytoplasm (GAPDH), nuclear envelope (Lamin B). p12 and p32, p37/39, p43/44, for cleaved forms of caspase-2, -9, and -8, respectively, with the corresponding molecular weight.

**Figure 5 cells-10-00852-f005:**
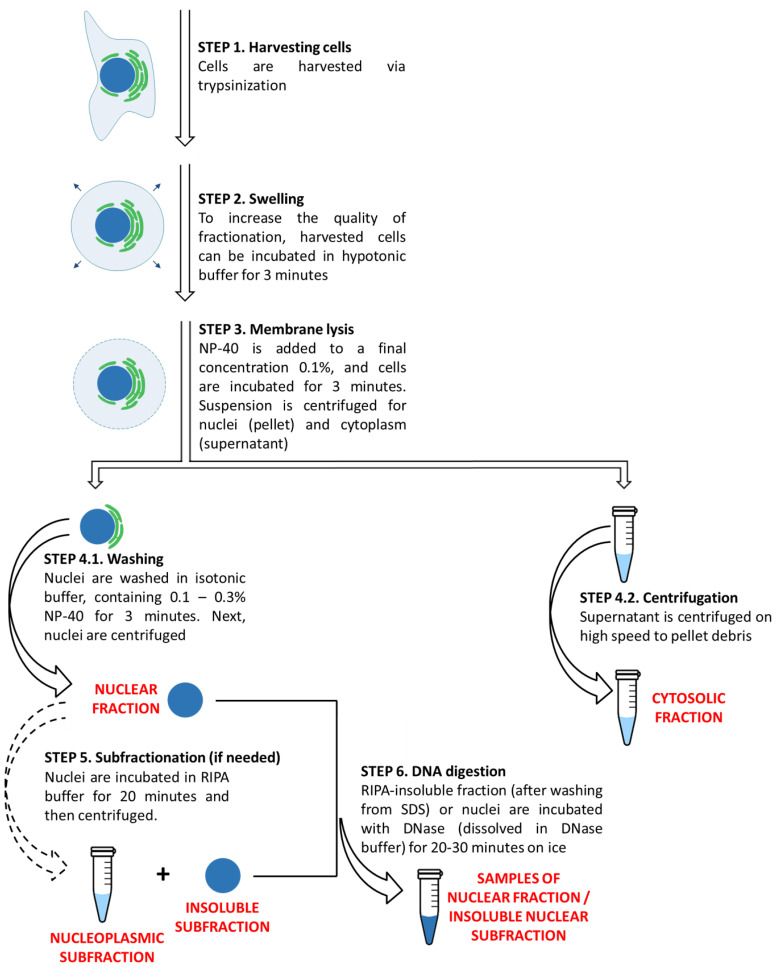
Graphical representation of the nucleus/cytoplasm fractionation protocol.

## Data Availability

Not applicable.

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
