# Peer review of "Simple and Efficient Protocol for Subcellular Fractionation of Normal and Apoptotic Cells"

_cells, 2021, doi:10.3390/cells10040852_

Round 1

Reviewer 1 Report

Review comments

This is an interesting manuscript that applies Lyse & Wash protocol for nucleus/cytoplasm fractionation in two different cell lines (Caov-4 and HeLa cells). Using the approach developed by the group, the author showed the nucleus and cytoplasm fraction can be successfully separated, and the nuclear fraction can be further divided into RIPA-soluble and -insoluble subfractions. Altogether, the methodology manuscript showed a highly effective method for nucleus/cytoplasm fractionation for both living and apoptotic cells. However, the L&W protocol only differed from the REAP method from the hypotonic solution usage and a few minutes durarion, dampened the novelty for the method. In addition, the group only test the method in two cancer cell lines, which limits the potential usage of the method. I strongly suggested that the author should provide more cells/tissues example including primary cells from mice, which would increase the insight value of the method.

Major comments:

  1. Figure 1 showed the western blotting and immune staining result of Caov-4 and Hela cell lines using different methods. However, in Figure 1A, it seemed the CytC and Lamin B were better in the iso group, rather than hyp group. In Figure 1C, there was no much difference betweenthe 0.1%, 0.2% and 0.3% of NP-40.
  2. The author showed the WB and immune staining use different method, however, how about the degradation condition of the protein/RNA using different method? Can this method also be applied for RNA-pull down of the nucleus proteins?

Minor points

Line 77:  What is the “REAP” short for?   The author should state the full name “R apid, E fficient A nd P ractical” for the first time use in the manuscript.

Reviewer 2 Report

The authors describe L&W methods to obtain a more efficient subcellular fractionation.

They also use this method to evaluate the fractionation of Apoptotic cells.

Personally, I have several doubts about the induction of apoptosis following the treatment with staurosporine and cis-platino. In figure 3, I don’t observe any induction of cell death in terms of Caspases activation. For all caspases, I observed the same pattern present in the ctrl cells. The antibodies used detected also the caspases ‘cleavage fragments. My suggestion is to explain this figure better, also because this point is important for the authors. If there is not induction of death is difficult to assume that this method discriminates the apoptotic cells.

Furthermore, they demonstrated the presence of some caspases in the nuclear fraction (albeit at low levels)

I think it is very difficult to observe these signals, moreover, for the expression of caspase 9 in Caov-4, in my opinion the same signals is present also in the nuclei of the control. I suggest to repeat this experiment or to improve the image quality.

Minor point

I suggest adding molecular weight of indicated proteins to all western blots

Round 2

Reviewer 2 Report

Dear authors,

thanks for the revision. Unluckily, I am still not convinced. I know how difficult it is to get a good cell fractionalization, so having a good method is of great scientific interest. I emphasize that the authors describe this method as a good strategy also to discriminate apoptotic cell. Considering that in some their experiments, the activation of apoptotic cell death is difficult to observe, I think it is not correct to propose this method for this evaluation.  

1- The authors replay that “… it is well-known that full cleavage of caspases is not needed to induce cell death”. Ok it is true, but my observation was “I observed the same pattern present in control cells”. I know that is difficult obtain a full caspases’ cleavage, but to say that caspases are activate, it is necessary observe a difference between pro-caspase (in ctrl) and activated caspase (in treatment samples).

e.g. Caspases shown in figure 3

2-  Concerning caspase 9, authors replay “we cannot exclude that in some cell lines C9 might translocate in the nucleus without apoptosis induction”. Again my question was about the active form of caspse9 and specifically that I am unable to observe any difference between treatment (where they say that C9 is active) and ctrl.  Also in ctrl cells C9 it moves in the nucleus in active form?

3- I observed the three figure provided in the response. Why don’t you add these additional panels to the manuscript? (e.g. as supplementary infos). However, even in Fig6 (for reviewers use only), there is no variation in procaspase 3 level between treated and not treated cells. Different is for proC9 (Caoc-4 experiment).

4- I asked to editor, to receive the whole western blot image to know how you have conducted the experiment. E.g. for caspase3, in the same panel you showed 12 western blot. Have all 11 antibodies been normalized with GAPDH? I think that this is not possible, and I supposed that you normalize some antibodies with a specific house-keeping and others with a different one. Please insert the respective house-keeping protein used for each antibody in the figure, and I suggest you to normalize the western blot using a specific image program (e.g. Image J)

5- I suggested to adding the molecular weight of indicated protein to all presented western blot images. The authors replay “… the figures were changed according to your suggestion”, where are these changes? I am based my question on the “peer-review R2” file that I downloaded from the site.

Author Response

Point-by-point response

Reviewer

Dear authors,

thanks for the revision. Unluckily, I am still not convinced. I know how difficult it is to get a good cell fractionalization, so having a good method is of great scientific interest. I emphasize that the authors describe this method as a good strategy also to discriminate apoptotic cell. Considering that in some their experiments, the activation of apoptotic cell death is difficult to observe, I think it is not correct to propose this method for this evaluation. 

Response:

Dear Sir/M-me,

We thank you again for your thorough work with our manuscript. Unfortunately, due to a technical error, there were no changes in figures in the file “peer-review R2”. We apologize for this and provide you with an actual version of our manuscript.

Additionally, please, find the answers to other comments below.

Comment 1.

The authors replay that “… it is well-known that full cleavage of caspases is not needed to induce cell death”. Ok it is true, but my observation was “I observed the same pattern present in control cells”. I know that is difficult obtain a full caspases’ cleavage, but to say that caspases are activate, it is necessary observe a difference between pro-caspase (in ctrl) and activated caspase (in treatment samples). e.g. Caspases shown in figure 3

Response:

Indeed, staining with Abs presented in this figure did not allow to clearly observe the difference between control and treated cells for some caspases. We substituted figure 3 with another one with better quality. Now, the difference in caspase cleavage is more evident.

Comment 2.

Concerning caspase 9, authors replay “we cannot exclude that in some cell lines C9 might translocate in the nucleus without apoptosis induction”. Again my question was about the active form of caspse9 and specifically that I am unable to observe any difference between treatment (where they say that C9 is active) and ctrl.  Also in ctrl cells C9 it moves in the nucleus in active form?

Response:

Please, see answer to Comment #1.

Comment 3.

I observed the three figure provided in the response. Why don’t you add these additional panels to the manuscript? (e.g. as supplementary infos). However, even in Fig6 (for reviewers use only), there is no variation in procaspase 3 level between treated and not treated cells. Different is for proC9 (Caoc-4 experiment).

Response:

These figures are a part of another project and that is why they are not to be used in this article.

The variation in caspase-3 cleavage can be seen by appearance of cleavage products – p17/19 of caspase-3. This staining was absent for Caov-4 cells in Fig. 6 (for the Reviewers use only). We included another figure into the revised version of the MS, in which caspase cleavage is more evident.

 Comment 4.

I asked to editor, to receive the whole western blot image to know how you have conducted the experiment. E.g. for caspase3, in the same panel you showed 12 western blot. Have all 11 antibodies been normalized with GAPDH? I think that this is not possible, and I supposed that you normalize some antibodies with a specific house-keeping and others with a different one. Please insert the respective house-keeping protein used for each antibody in the figure, and I suggest you to normalize the western blot using a specific image program (e.g. Image J)

Response:

Based on your request we provided Editors with all the raw files. As for experimental procedures in figure 3, in brief, there were two Western blots with the same samples for each cell line. Next, the membranes were cut into parts depending on molecular weights of proteins of interest. Each of two membranes was stained for GAPDH. Of note, in these experiments GAPDH is not the only marker protein. Other proteins, such as Cytochrome c and Erp-29, are controls too which are used as markers for specific intracellular compartments.

Comment 5.

I suggested to adding the molecular weight of indicated protein to all presented western blot images. The authors replay “… the figures were changed according to your suggestion”, where are these changes? I am based my question on the “peer-review R2” file that I downloaded from the site.

Response:

Again, we are very sorry for this technical error. We pasted new figures in a “track changes” mode. However, they were, indeed, absent in the “peer-review R2” file. We hope this error will not occur this time.